# Heterogeneity of Mismatch Repair Status and Microsatellite Instability between Primary Tumour and Metastasis and Its Implications for Immunotherapy in Colorectal Cancers

**DOI:** 10.3390/ijms23084427

**Published:** 2022-04-17

**Authors:** Camille Evrard, Stéphane Messina, David Sefrioui, Éric Frouin, Marie-Luce Auriault, Romain Chautard, Aziz Zaanan, Marion Jaffrelot, Christelle De La Fouchardière, Thomas Aparicio, Romain Coriat, Julie Godet, Christine Silvain, Violaine Randrian, Jean-Christophe Sabourin, Rosine Guimbaud, Elodie Miquelestorena-Standley, Thierry Lecomte, Valérie Moulin, Lucie Karayan-Tapon, Gaëlle Tachon, David Tougeron

**Affiliations:** 1Medical Oncology Department, Poitiers University Hospital, 86000 Poitiers, France; camille.evrard@chu-poitiers.fr; 2Hepato-Gastroenterology Department, Poitiers University Hospital, 86000 Poitiers, France; steph.messina@outlook.com (S.M.); christine.silvain@chu-poitiers.fr (C.S.); violaine.randrian@chu-poitiers.fr (V.R.); 3UNIROUEN, Inserm 1245, Group IRON, Normandie University, Gastroenterology Department, Rouen University Hospital, 76000 Rouen, France; david.sefrioui@chu-rouen.fr; 4Poitiers University Hospital, 86000 Poitiers, France; eric.frouin@chu-poitiers.fr (É.F.); lucie.karayan-tapon@chu-poitiers.fr (L.K.-T.); gaelle.tachon@chu-poitiers.fr (G.T.); 5Department of Pathology, Poitiers University Hospital, 86000 Poitiers, France; julie.godet@chu-poitiers.fr; 6Department of Pathology, La Rochelle Hospital, 17000 La Rochelle, France; marie-luce.auriault@ght-atlantique17.fr; 7Gastroenterology and Digestive Oncology Department, Tours University Hospital, 37000 Tours, France; romain.chautard@gmail.com (R.C.); thierry.lecomte@univ-tours.fr (T.L.); 8Gastroenterology and Digestive Oncology Department, Georges Pompidou European Hospital, Assistance Publique des Hôpitaux de Paris (APHP), 75015 Paris, France; aziz.zaanan@aphp.fr; 9Gastroenterology Department, Toulouse Rangueil University Hospital, 31000 Toulouse, France; jaffrelot.m@chu-toulouse.fr (M.J.); guimbaud.r@chu-toulouse.fr (R.G.); 10Medical Oncology Department, Centre Léon Bérard, 69000 Lyon, France; christelle.delafouchardiere@lyon.unicancer.fr; 11Gastroenterology and Digestive Oncology Department, Saint Louis Hospital, Assistance Publique des Hôpitaux de Paris (APHP), 75012 Paris, France; thomas.aparicio@aphp.fr; 12Gastroenterology and Digestive Oncology Department, Cochin Hospital, Assistance Publique des Hôpitaux de Paris (APHP), 75012 Paris, France; romain.coriat@aphp.fr; 13Inflammation, Tissus Épithéliaux and Cytokines Laboratory, EA 4331, Poitiers University, 86000 Poitiers, France; 14Department of Pathology, INSERM U1245, Rouen University Hospital, 76000 Rouen, France; jean-christophe.sabourin@chu-rouen.fr; 15Department of Pathology, Tours University Hospital, 37000 Tours, France; elodie.standley@univ-tours.fr; 16EA 4245, Transplantation, Immunologie, Inflammation, Tours University, 37000 Tours, France; 17Inserm UMR 1069, Nutrition, Croissance et Cancer, Tours University, 37000 Tours, France; 18Medical Oncology Department, La Rochelle Hospital, 17000 La Rochelle, France; valerie.moulin@ght-atlantique17.fr; 19INSERM Laboratoire de Neurosciences Expérimentales et Cliniques, Poitiers University, 86000 Poitiers, France; 20Cancer Biology Department, Poitiers University Hospital, 86000 Poitiers, France

**Keywords:** colorectal cancer, microsatellite instability, deficient mismatch repair, immunohistochemistry, inter-tumoral heterogeneity

## Abstract

Deficient mismatch repair system (dMMR)/microsatellite instability (MSI) is found in about 5% of metastatic colorectal cancers (mCRCs) with a major therapeutic impact for immune checkpoint inhibitor (ICI) use. We conducted a multicentre study including all consecutive patients with a dMMR/MSI mCRC. MSI status was determined using the Pentaplex panel and expression of the four MMR proteins was evaluated by immunohistochemistry (IHC). The primary endpoint was the rate of discordance of dMMR/MSI status between primary tumours and paired metastases. We included 99 patients with a dMMR/MSI primary CRC and 117 paired metastases. Only four discrepancies (3.4%) with a dMMR/MSI primary CRC and a pMMR/MSS metastasis were initially identified and reviewed by expert pathologists and molecular biologists. Two cases were false discrepancies due to human or technical errors. One discordant case could not be confirmed due to the low level of tumour cells. The last case had a confirmed discrepancy with a dMMR/MSI primary CRC and a pMMR/MSS peritoneal metastasis. Our study demonstrated a high concordance rate of dMMR/MSI status between primary CRCs and their metastases. The analysis of one sample, either from the primary tumour or metastasis, with consistent dMMR and MSI status seems to be sufficient prior to treatment with ICI.

## 1. Introduction

Colorectal cancer (CRC) is an important public health issue, with approximately 500,000 new cases and 700,000 deaths worldwide each year [1]. The three major pathways involved in CRC carcinogenesis are chromosomal instability (75% of the CRCs), microsatellite instability (MSI) (15%), and a CpG island methylator phenotype (CIMP) (25%) [2]. These pathways are not mutually exclusive, especially insofar as there is an overlap between MSI phenotype and CIMP.

The mismatch repair (MMR) system identifies and corrects DNA mismatches missed by DNA polymerase during replication. The MMR system consists of four major proteins: MLH1, MSH2, MSH6, and PMS2. Deficiency of the MMR system (dMMR) is responsible for the MSI phenotype observed in tumors in contrast to proficient MMR systems (pMMR) with microsatellite stable (MSS) tumors. MMR deficiency results from either germline mutation of one of the MMR genes (Lynch syndrome, 20% of dMMR/MSI CRCs), or somatic inactivation of one of the MMR genes, mostly *MLH1* by hypermethylation of its promoter (sporadic cases, 80% of dMMR/MSI CRCs) [3].

MSI results in an accumulation of frameshift mutations at coding microsatellites leading to the generation of mutated and truncated proteins, which play a major role in dMMR/MSI CRC carcinogenesis and in the generation of immunogenic neo-antigens [4]. Indeed, immune checkpoint inhibitors (ICI) are highly effective in patients with dMMR/MSI metastatic CRC (mCRC), correlated with the high tumor mutational burden and the high neo-antigen burden of these tumors [5]. The phase III KEYNOTE-177 trial showed that the progression-free survival (PFS) of patients with dMMR/MSI advanced CRC treated with the anti-programmed death-1 (PD-1) Pembrolizumab was double the PFS observed with chemotherapy in the first-line setting (16.5 months versus 8.2 months) [6]. In addition to its therapeutic impact, the accurate determination of dMMR/MSI status is critical for Lynch syndrome (LS) screening.

Two main techniques are currently used to determine dMMR/MSI status: immunohistochemistry (IHC), which detects loss of MMR protein expression in the tumour (dMMR status), and molecular testing, which determines the fragment size of microsatellite loci on tumor DNA (MSI status). Both techniques are recognized worldwide, but recent studies comparing the two most robust techniques, IHC of the four MMR proteins (MLH1, MSH2, MSH6, PMS2) and the molecular MSI test using the Pentaplex panel, showed a 1–5% discordance rate [7,8,9]. Since ICIs are not effective in pMMR/MSS mCRC, the performance of both MMR IHC and MSI tests is now recommended before ICI initiation in recent guidelines, since the performance of only one test increases the risk of mistakenly considering a tumor as dMMR/MSI [10,11]. In addition, a recent report suggests rare cases of intra-tumoral heterogeneity of dMMR/MSI status in primary CRCs and either the dMMR/MSI sub-clone or the pMMR/MSS sub-clone could yield metastases [12].

Inter-tumoral heterogeneity, between primary CRC and metastases, of dMMR/MSI status is also a subject of debate with recent studies showing contradictory results. Haraldsdottir et al. reported a concordance rate of 100% between the primary tumour and metastases using MMR IHC alone in a cohort of 24 mCRC patients [13]. By contrast, other small series have shown a concordance rate ranging from 42% to 87.5% using either MMR IHC or an MSI test, but never both tests and without review of cases by experts [14,15]. Only one study performed a comparative analysis of 48 dMMR/MSI primary CRCs with their paired metastases using both tests. MSI analysis was performed using five microsatellite markers (BAT-25, BAT-26, D2S123, D5S346, and D17S250) and MMR IHC was performed with the IHC of the four MMR proteins. Results showed an 18.7% discrepancy (n = 9/48) [16]. All discrepancies were observed for peritoneal metastases and ovarian metastases (n = 5/20 and n = 4/4, respectively). It is noteworthy that among these nine discrepancies, six metastases were pMMR/MSS while three metastases were dMMR/MSS (discordance between the MSI test and MMR IHC). In addition, despite the fact that multiple known factors cause discrepancies, e.g., the expertise of pathologists, quality of tissue sampling, quality of tissue fixation and staining, and low rates of tumor cells, no case has been extensively reviewed by authors [8,17,18]. Indeed, some cases could be false discrepancies since peritoneal metastases are frequently associated with few tumor cells, especially after chemotherapy [19].

As the inter-tumoral heterogeneity of dMMR/MSI status could be a major concern for ICI use and as only small series with conflicting results are available, we aimed to analyze a larger series of dMMR/MSI mCRCs to determine the frequency of these discrepancies between primary tumors and paired metastases.

## 2. Results

### 2.1. Study Population

The median age at CRC diagnosis was 59.0 years and most patients were male (52.5%) (Table 1). Among the 99 patients, 117 metastatic samples were available, 55 were synchronous and 62 metachronous to the primary tumor (Figure 1).

### 2.2. Metastatic Samples

The number of metastases ranged from 1 to 3 per patient. All in all, 117 metastatic samples were analyzed. The median delay between primary tumor diagnosis and metastasis diagnosis was 7.0 months. Most metastases were metachronous (53.0%). The most frequent metastatic sites were the peritoneum (46.2%), the liver (23.9%), and the lymph nodes (12.0%) (Table 2).

Altogether, 34.2% patients received chemotherapy before MMR IHC or MSI tests on their metastases. MMR IHC and MSI tests were performed on 95.7% and 88.0% of metastatic samples, respectively. No discordance between MMR and MSI status in primary tumors or metastases was observed (no dMMR/MSS or pMMR/MSI case). *RAS* and *BRAF^V600E^* status were available in 53.0% and 61.4% in primary tumors and paired metastases, respectively. No discordance of *RAS* or *BRAF^V600E^* status was observed between primary tumors and paired metastases. Among primary tumors and paired metastases with the *MLH1* promoter methylation test available (66.0%), no discordance was observed between primary tumors and metastases.

### 2.3. Concordance of MMR Immunohistochemistry and MSI Status between Primary Tumors and Paired Metastases

All in all, 91.2% (n = 196/215) of tumor samples were compared by MMR IHC and the MSI test in both primary tumors and metastases. All available primary tumors had both tests and 83.8% of metastases had both tests (95.7% of MMR IHC (n = 112/117) and 88.0% of MSI tests (n = 103/117)).

The dMMR/MSI status of the primary tumor was consistent with the MMR and/or MSI status of paired metastasis(es) in 96.6% of patients, meaning there were four cases with discordance (3.4%). The four discordant cases had both MMR IHC and MSI tests in metastases and were initially classified as pMMR/MSS. For patients with multiple metastatic samples, no discordance between the primary tumors and paired metastases was detected. The four discordant cases had only one metastatic sample available to perform MMR IHC and MSI tests. In dMMR primary cancers with dMMR metastases, no discordance concerning the kind of MMR proteins expression loss (MSH2, MSH6, MLH1, and/or PMS2) between primary tumors and paired metastases was observed.

For all cases with discordant results between MSI and MMR IHC tests, re-examination of the molecular MSI profile and the MMR IHC staining was performed by experts. Among the four discrepancies, two had a precise explanation after the review of pathological samples, MMR IHC, and MSI tests.

### 2.4. Description of the Four Cases with Discordant MMR IHC and MSI Status between Primary Tumors and Paired Metastases

The first discordant case (case 1) was a 73-year-old man who presented a stage II dMMR/MSI right colon treated by surgery in 2005. MMR IHC of the primary tumor showed the loss of MLH1 and PMS2 expressions. Nine months after surgery of the right colon cancer, the patient developed lung metastases, which were resected (lower left lobectomy) and showed pMMR/MSS status. After patient file verification and proofreading, this patient had two distinct metachronous primary colon cancers, a dMMR/MSI, *BRAF*-mutated and hypermethylated *MLH1* promoter, right colon cancer in 2005, and in 2003 a pMMR/MSS, *BRAF* wild-type, left-sided colon cancer stage II treated by surgery (Figure 2). Indeed, it was probably a false discrepancy since the lung metastases may have come from the pMMR/MSS left-sided colon cancer.

The second discordant case (case 2) was a 70-year-old woman who had a synchronous resection of a right colon cancer and a peritoneal lesion initially considered as a metastatic lesion. The primary lesion showed a dMMR (MLH1 and PMS2 loss) and MSI status. Analysis of the synchronous peritoneal lesion showed no MMR protein loss (pMMR) and MSS status. After re-examination of the peritoneal lesion by an expert pathologist, it was concluded that the sample had no tumor cells but only atypical mesothelial hyperplasia, thereby explaining the pMMR/MSS status (no tumor cells) and false discordance between the primary tumor and peritoneal lesion.

The third discordant case (case 3) was an 83-year-old woman who underwent a synchronous resection of a right colon adenocarcinoma with a biopsy of unresectable peritoneal carcinomatosis. Analyses of the non-mucinous primary tumor found a dMMR (loss of MLH1 and PMS2) and MSI status. Review of the pathological sample confirmed synchronous peritoneal carcinomatosis. The tumor DNA extracted from the peritoneal carcinomatosis showed MSS status with no instability of the five microsatellites analyzed. The peritoneal sample was supposed to contain more than 50% of tumor cells but we succeeded in extracting only 0.1 ng/μL of DNA. The low concentration of DNA was not sufficient to provide unquestionable molecular results. Unfortunately, there was no remaining tumor tissue to perform MMR IHC or a new MSI test. This case could represent true inter-tumoral heterogeneity of MMR/MSI status between the primary lesion and synchronous peritoneal carcinomatosis or, more likely, a false discrepancy due to the limitation in tumor sample availability.

The fourth discordant case (case 4) was a 75-year-old woman, who underwent synchronous surgery of non-mucinous right colon cancer and peritoneal carcinomatosis. The analysis of the primary tumor with 80% of tumor cells showed a loss of MLH1 and PMS2 (dMMR) with MSI status and a *BRAF^V600E^* mutation. The analysis of the peritoneal carcinomatosis found a pMMR and MSS status and no *BRAF^V600E^* mutation. The pathological characteristics of the primary tumor and the metastasis were similar. An extensive review on multiple tumor blocks and reanalysis of the MMR IHC and MSI status were performed and did not explain the discrepancy, in particular, no subclone was detected (Figure 3). The different imaging performed including a CT-scan and the follow-up of this patient did not show a second cancer. This last case was the only one with confirmed inter-tumoral heterogeneity of MMR/MSI status between primary tumor and metastasis.

Finally, among the four discordant cases between primary tumors and metastases, two were false discordances (cases 1 and 2), one was questionable (case 3), and only one seemed to be a true discordance (case 4).

## 3. Discussion

Our study aimed to analyze a large-scale series of dMMR/MSI mCRCs to determine the frequency of MMR IHC and MSI discrepancies between primary tumors and paired metastases. To our knowledge, this is the largest series, with 98 primary dMMR/MSI CRC and 117 paired metastases paired samples, which analyzed dMMR/MSI mCRC both by MMR IHC (four MMR proteins) and MSI (Pentaplex) in contrast to previous smaller series using only one technique. In addition, all discordant cases have been proofread by expert pathologists and molecular biologists limiting human and technical errors. We demonstrated a high concordance of dMMR/MSI status between dMMR/MSI primary tumors and their metastases in 99 patients presenting a dMMR/MSI primary CRCs and 117 paired metastases. An initial discrepancy was detected for only four patients (3.4%) with dMMR/MSI primary tumors and pMMR/MSS metastases. Two cases were explained with human or technical limits: the first one with no tumor cells in the sample and the second one with too low a concentration of tumor DNA. One case was due to a metachronous pMMR/MSS tumor, in addition to dMMR/MSI colon cancer. Finally, only one case remained discordant. In light of our results, if inter-tumoral heterogeneity in dMMR/MSI status exists in CRC, it is infrequent. This result is in accordance with daily clinical practice, where testing of metastatic lesions from dMMR/MSI primary mCRCs are also dMMR/MSI.

The exact reasons for potential discrepancies in dMMR/MSI status between primary tumors and metastases are for the moment unknown. One hypothesis would be that they are the consequences of tumor subclones that co-exist within the primary CRC (dMMR/MSI and pMMR/MSS subclones), as was recently reported [20,21]. These subclones could have major therapeutic impacts for treatment with ICI since it has been shown that only dMMR/MSI mCRC benefit from immunotherapy and not pMMR and/or MSS mCRC [22]. Indeed, cases with a misclassification of MMR IHC and/or the MSI test have been associated with a resistance to ICI [23]. To our knowledge, no study has evaluated whether minor pMMR/MSS subclones can explain primary or secondary resistances to ICI of dMMR/MSI mCRC.

A study on 24 patients showed 100% of concordance of MMR IHC between primary tumors and paired metastases but the population was small and authors used only MMR IHC without molecular testing to confirm MSI status [13]. Conversely, another study showed a lower concordance between primary tumors and metastases (47%) using only IHC and this study included only seven dMMR CRCs [14]. It is worth noting that most of the discrepancies found in the literature have concerned peritoneal carcinomatosis. Indeed, Wen Zhuo He et al. recently reported nine cases of discrepancies between nine dMMR/MSI primary CRC and paired metastases, five MSS peritoneal metastases and four MSS ovarian metastases [16]. However, two of these cases presented dMMR profiles when IHC was performed on the metastatic tissue and are therefore likely to be false positive due to technical errors. The higher rate of MMR/MSI discrepancy between primary CRC and peritoneal carcinomatosis could also be explained by the fact that tumor samples are frequently poor, with a low amount of tumor cells, which can prevent MSI and MMR IHC tests from analyzing properly. To obtain reliable results, more than 20% of tumor cells are necessary for MSI testing [17]. For MMR IHC testing, the internal positive control is fundamental to avoid misclassification, especially in the cases with high mucinous components, which compose a very high amount of tumor material. Tissue from peritoneal carcinomatosis usually combines both drawbacks, with few tumor cells, and a high mucinous component. In the Wen Zhuo He et al. study there is no information concerning any of these points. The population was small (n = 46) and there was no proofreading by pathology and molecular biology experts. In our study, the two discordant cases (cases 3 and 4) concerned peritoneal carcinomatosis with no mucinous component, one with few tumor cells for MMR IHC and a low amount of tumor DNA for MSI testing, and the second with a high amount of tumor cells and unexplained discrepancy. Regarding all these results, in case of the discrepancy between the peritoneal carcinomatosis and primary tumor, it seems necessary to reassess pathological and molecular results and to verify the amount of tumor cells in the sample.

In addition, rare cases of discrepancy of MMR/MSI status between two tumor samples have been attributed to radiotherapy or chemotherapy [24]. In the Wen Zhuo He et al. study, there is no information concerning this point. In our series, discrepancies were observed in cases with synchronous metastases and no neo-adjuvant treatment. Moreover, it is important to verify the pathological nature of the metastasis and its concordance with the primary tumor, as well as the absence of multiple primary tumors, as this explains two discrepancies in our series. Finally, it is of major importance to use both techniques (MMR IHC and MSI tests) on each tumor sample available (primary tumor and paired metastases). Even if in our work no discordance between MSI and MMR IHC tests was identified, it may exist in about 1–3% of the CRC [8,25].

In order to avoid discrepancies of MMR/MSI status between primary tumors and metastases, a recent study used plasma based MSI detection (bMSI) developing an algorithm that included 100 microsatellite markers. In comparison with the PCR in the tissue, bMSI displayed a sensitivity of 82.5% and a specificity of 96.2% [25]. The researchers performed a clinical validation on 60 patients with advanced or metastatic gastrointestinal cancer treated with anti-PD-1/anti-PD-L1 monotherapy. MSI/MSS tumor status was unfortunately unknown, but 31 were bMSI positive. Patients with positive bMSI results have better overall survival and PFS as compared to bMSS. More data are necessary to determine whether bMSI is more accurate than tumor PCR or MMR IHC to determine the sensitivity to ICIs.

The strength of our multicentric study is the size of our cohort. In addition, most tumor samples had a combined analysis of MMR IHC and the MSI test, which is in accordance with ESMO guidelines before treatment with ICI [10]. Moreover, all MMR IHC and MSI tests were performed by expert cancer care teams and discordant cases have been blind proofread by expert pathologists and molecular biologists. After these controls, very few discordances remained.

The main limitation of our study is its retrospective nature. It would have been interesting to analyze the concordance between pMMR/MSS primary CRC and paired metastases in order to determine whether pMMR/MSS tumors may present dMMR/MSI metastasis during progression and under treatment. However, this was not the objective of our study. In addition, this was hardly feasible, as it would have required the analysis of many tumors to find very few or no discordant cases (pMMR/MSS primary tumor with dMMR/MSI metastasis by somatic inactivation). Plasma-based microsatellite instability detection could help to perform this kind of study in the future. Besides, this type of discordance is unlikely, as it is admitted that pMMR tumors, and therefore pMMR subclones, have more metastatic potential than dMMR tumors/subclones [26]. Thus, testing of the metastatic lesions does not seem more appropriate than testing of the primary tumor given that previous series and our study do not support changes in MMR/MSI status during metastatic dissemination.

To our knowledge, this is the first multicentric study with almost 100 patients to compare concordance of dMMR/MSI status between primary CRC and their paired metastases. Our series demonstrates a high concordance of dMMR/MSI status between primary tumor and paired metastases, with only one discordance concerning a peritoneal carcinomatosis. While other studies are necessary to confirm our results, inter-tumoral heterogeneity of dMMR/MSI status seems extremely rare. Based on these results, we do not recommend the systematic testing of dMMR/MSI status on metastases to confirm the results obtained on primary tumors, unless the results are equivocal. A metastasis biopsy is not mandatory to determine dMMR/MSI status before starting treatment with an ICI if a primary tumor is available. Nevertheless, as recommended by ESMO guidelines, both MMR IHC and MSI tests must be performed before treating a patient with ICI, the objective being to avoid false positive cases. Indeed, in the case of a discrepancy between MMR IHC and MSI tests on primary tumors, testing of metastatic samples, if available, could solve the problem. It might then be interesting to evaluate MSI status directly on circulating tumor DNA, which may better reflect tumor heterogeneity and may offer sequential screening of tumor subclones under ICI treatment to achieve the early detection of patients at a high risk of progression.

## 4. Materials and Methods

### 4.1. Study Population

In this retrospective multicentre study, we included all consecutive patients with a dMMR/MSI mCRC diagnosed between 2007 and 2019. dMMR/MSI CRCs were identified using the local clinical database, pathology department and/or cancer biology department databases, as previously described [27]. Inclusion criteria were histologically proven dMMR and/or MSI CRC with at least one metastasis for which tumor material was available. Non-inclusion criteria were not available for MMR IHC and MSI results on metastatic samples. The study was approved by the ethics committee “*Comité de Protection des Personnes Ouest III*” (DC-2008-565). As we performed a retrospective study and most of the patients had died, no written consent was required. This study has been conducted in accordance with current French law and with the ethical principles of the Helsinki Declaration of 1975 and its subsequent revisions.

We initially identified 127 patients eligible for the study (Figure 1). All but one patient had a dMMR or MSI primary tumors and at least one metastatic sample available. For one patient, we had two different metastatic samples (dMMR/MSI) but no primary tumor available. We excluded 28 patients because there was no residual tumor left on metastatic samples for MSI and MMR IHC tests. Finally, 99 patients were analyzed, including 98 patients with MMR IHC and/or MSI tests available on both primary and metastatic samples. For 15 patients, 2 or 3 metastatic samples from different sites were available (Figure 1). All in all, 215 tumor samples were analyzed (98 primary tumors and 117 metastatic samples).

### 4.2. MMR Immunohistochemistry and MSI Tests

MMR status was determined by DNA MSI testing (Pentaplex panel) and/or analysis of the four MMR protein expressions by IHC. Deficient MMR status was defined by nuclear loss of at least one MMR protein (MLH1, MSH2, MSH6, or PMS2) in tumour cells while normal cells remained stained, using a BenchMark XT device (Ventana Medical Systems^®^, Tucson, AZ, USA). The antibodies used were anti-MLH1 (clone M1 Ventana^®^, Tucson, AZ, USA, kit Optiview^®^, Tucson, AZ, USA for revelation); anti-MSH2 (clone G219-1129 Ventana^®^, kit Optiview^®^ for revelation); anti-MSH6 (clone 44BD Biosciences^®^ San Jose, CA, USA, kit Ultraview^®^ for revelation), and anti-PMS2 (clone EPR 3947 Ventana^®^, ready for use; kit Optiview^®^ for revelation with amplification).

We used a DNeasy Blood and Tissue DNA isolation kit^®^ (Qiagen^®^, Hilden, Germany) to extract DNA from formalin-fixed paraffin-embedded (FFPE) tissue. MSI was assessed with the Pentaplex mononucleotide repeat panel (BAT-25, BAT-26, NR-21, NR-22, and NR-24) using the Promega^®^ MSI assay (Promega^®^, Madison, WI, USA) and analyzed on an ABI PRISM 3100 genetic analyser (Applied Biosystems^®^, Foster City, CA, USA), as previously described [8]. From extracted tumor DNA, MSI was defined by the instability of at least three microsatellite markers.

MMR IHC and MSI tests, whenever possible, were performed on a sample prior to any chemotherapy or radiotherapy. For all cases with discordant results between MSI and MMR IHC tests and/or discrepancies between primary tumors and paired metastases, re-examination of the molecular MSI profile and the MMR IHC staining were performed by experts.

### 4.3. Patient and Tumor Characteristics

Patients (age, gender, personal and family medical history of cancer, germline MMR testing results), tumors (date of diagnosis, primary tumor site, grade, TNM stage at diagnosis, number, and sites of metastases) and treatment characteristics were collected, as were *RAS*, *BRAF* and *MLH1* promoter methylation status. The *MLH1* promoter methylation test was performed only in tumors with MLH1 and/or PMS2 loss at MMR IHC. The determination of sporadic dMMR/MSI versus suspected LS cases was based on MMR protein expression, family history, *BRAF* status, and *MLH1* promoter methylation status, as previously described [27,28].

### 4.4. Statistical Analyses

Continuous variables were described with median and range and qualitative variables with frequency and percentages. Median delays between primary tumors and metastasis diagnosis were calculated. The primary endpoint was the rate of discordance of 

dMMR or MSI status between primary tumors and paired metastases. The secondary endpoint was to identify predictive factors of these discordances if they existed.

## Figures and Tables

**Figure 1 ijms-23-04427-f001:**
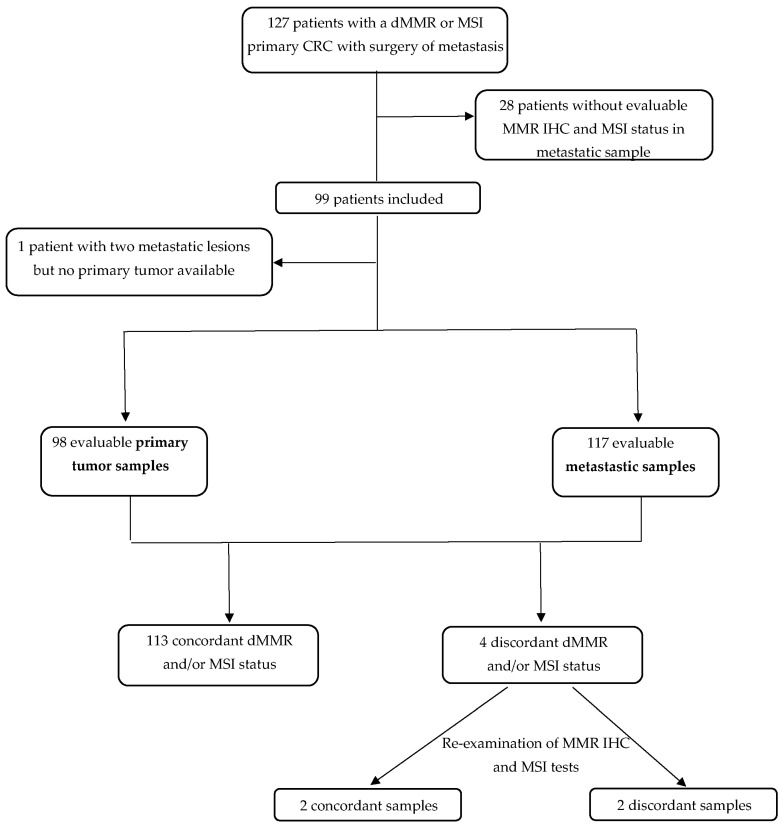
Flow chart. CRC: colorectal cancer; IHC: immunohistochemistry; MMR: mismatch repair; dMMR: deficient mismatch repair; MSI: microsatellite instability; MSS: microsatellite stability.

**Figure 2 ijms-23-04427-f002:**
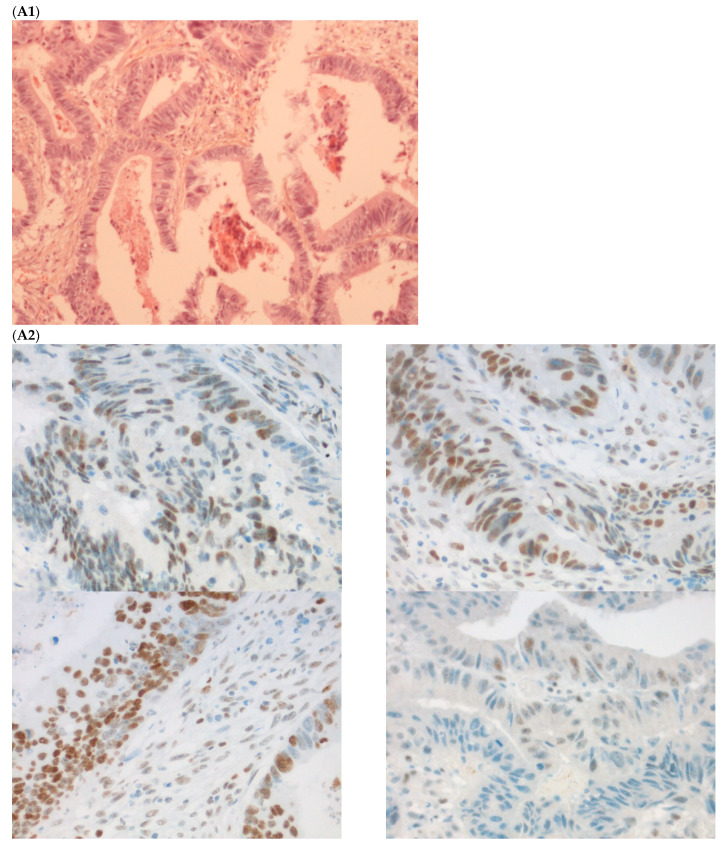
Pathological examination and MMR immunohistochemistry of the case number 1. (**A**) Primary left-sided colon cancer in 2003 with pMMR status ((**A1**), hematoxylin and eosin stain, 100× magnification) with an expression of the four MMR proteins (400× magnification; (**A2**) upper left: MLH1, upper right: MSH2, bottom left: MSH6 and bottom right: PMS2). (**B**). Primary right-sided colon cancer in 2005 with dMMR status ((**B1**), hematoxylin and eosin stain, 100× magnification) with a loss of MLH1 and PMS2 expression ((**B2**), upper left: MLH1 and upper right: PMS2) and MSH2 and MSH6 expression (bottom left: MSH2 and bottom right: MSH6). (**C**) Lung metastasis with pMMR status ((**C1**), hematoxylin and eosin stain, 100× magnification) with an expression of the four MMR proteins (400× magnification; (**C2**) upper left: MLH1, upper right: MSH2, bottom left: MSH6 and bottom right: PMS2). IHC: immunohistochemistry; MMR: mismatch repair.

**Figure 3 ijms-23-04427-f003:**
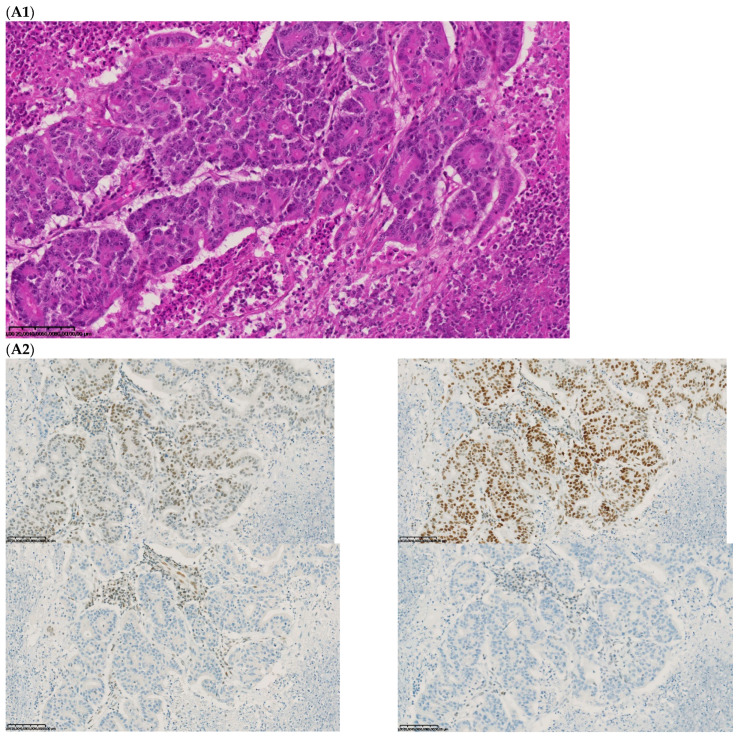
Pathological examination and MMR immunohistochemistry of the discordant case number 4. (**A**) Primary tumor with dMMR status ((**A1**), hematoxylin and eosin stain, 200× magnification) with an expression of MSH2 (**A2**, **up left**) and MSH6 (**A2**, **up right**) protein expression and a loss of MLH1 (**A2**, **down left**) and PMS2 (**A2**, **down right**). (**B**) Metastatic synchronous peritoneal carcinomatosis with a pMMR status ((**B1**), hematoxylin and eosin stain, 200× magnification) with an expression of MSH6 ((**B2**, **up left**), 200× magnification), MSH2 ((**B2**, **up right**), 200× magnification), MLH1 ((**B2**, **down left**), 200× magnification), and PMS2( (**B2**, **down right**), 300× magnification). IHC: immunohistochemistry; MMR: mismatch repair.

**Table 1 ijms-23-04427-t001:** Patient and tumor characteristics.

Characteristics (n = 99)	n (%)
**Median age** (years, range)	59.0 [17, 98]
**Gender (n = 99)**	
Female	47 (47.5%)
Male	52 (52.5%)
**TNM stage at diagnosis (n = 99)**	
I–II	14 (14.2%)
III	31 (31.3%)
IV	54 (54.5%)
**Primary tumour site (n = 98 *)**	
Right colon	65 (66.3%)
Left colon	28 (28.6%)
Rectum	5 (5.1%)
**Grade (n = 98 *)**	
Poorly differentiated	38 (39.6%)
Moderately differentiated	43 (44.8%)
Well differentiated	15 (15.6%)
Missing	2
***RAS* status on primary tumors (n = 98 *)**	
Mutated	20 (23.3%)
Wild-type	66 (76.7%)
Missing	12
**B*RAF^V600E^* status on primary tumors (n = 98 *)**	
Mutated	36 (38.3%)
Wild-type	58 (61.7%)
Missing	4
***MLH1* promoter hypermethylation (n = 75 **)**	
Yes	46 (70.8%)
No	19 (29.2%)
Missing	10
**Lynch syndrome or sporadic cases (n = 99)**	
Proven Lynch syndrome (MMR gene mutation)	20 (20.8%)
Suspected Lynch syndrome	21 (21.9%)
Sporadic case	55 (57.3%)
Missing	3

* One patient has no primary tumor sample available. ** Only tumors with MLH1 and/or PMS2 loss were tested for *MLH1* promoter hypermethylation. TNM: tumor, node, metastasis; MMR: mismatch repair; IHC: immunohistochemistry. Most primary tumors were right-sided (66.3%), poorly or moderately differentiated (84.4%) and stage III or IV at diagnosis (85.8%). All primary tumors were dMMR and had MSI status with no discordance. Most tests were performed before any treatment (92.9%). Among primary tumors, most presented a loss of MLH1 and PMS2 (70.4%) or a loss of MSH2 and MSH6 (17.3%) detected by MMR IHC (Table 2). *RAS* and *BRAF^V600E^* mutations were observed in 23.3% and 38.3%, respectively. All in all, 57.3% were sporadic cases, 42.7% were suspected or proven LS.

**Table 2 ijms-23-04427-t002:** MMR immunochemistry and MSI tests on primary tumors and paired metastases.

Characteristics (n, %)	Primary Tumors (n = 98)	Metastases (n = 117)
**Chemotherapy or radiochemotherapy before testing**		
Yes	7 (7.1%)	39 (34.2%)
No	91 (92.9%)	75 (65.8%)
Missing	0	3
**Synchronous/metachronous**	-	55 (47.0%)/62 (53.0%)
**Site of metastases**	-	
Peritoneum	-	54 (46.2%)
Liver	-	28 (23.9%)
Lymph nodes	-	14 (12.0%)
Lung	-	6 (5.1%)
Others	-	15 (12.8%)
**MMR IHC status**		
Loss of MLH1 and PMS2 expression	69 (70.4%)	72 (61.5%)
Loss of MSH2 and MSH6 expression	17 (17.3%)	21 (18.7%)
Isolated loss of PMS2 expression	6 (6.1%)	10 (8.9%)
Isolated loss of MSH6 expression	2 (2.1%)	1 (0.9%)
Others	4 (4.1%)	4 (3.6%)
pMMR	0	4 (3.6%)
Missing	0	5
**MSI test**		
MSI	98 (100%)	99 (96.1%)
MSS	0	4 (3.9%)
Missing	0	14
**MMR IHC and MSI tests **	98/98 (100%)	98/117 (83.8%)

IHC: immunohistochemistry; MMR: mismatch repair; MSI: microsatellite instability; pMMR: proficient MMR.

## Data Availability

The data that support the findings of this study are available on request from the corresponding author.

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
