# Peer review of "Heterogeneity of Mismatch Repair Status and Microsatellite Instability between Primary Tumour and Metastasis and Its Implications for Immunotherapy in Colorectal Cancers"

_ijms, 2022, doi:10.3390/ijms23084427_

Round 1

Reviewer 1 Report

Dear authors I've read with interest your paper. I think it is well written, clear and precise. The methods are well described and the results clearly stated. The discussion is well balanced.
I only ask for minor revisions:
-table 2, last line: unclear
-discussion: too many times you repeat that your study is the largest series
-discussion: I would emphasize that your study confirms the daily clinical practice

Author Response

REVIEWER #1

Dear authors I've read with interest your paper. I think it is well written, clear and precise. The methods are well described and the results clearly stated. The discussion is well balanced.

I only ask for minor revisions:

- table 2, last line: unclear

We thank the reviewer for highlighting the importance of our manuscript. Thank you for this careful reading, this last line of table 2 is incorrect and has been deleted.

- discussion: too many times you repeat that your study is the largest series

We have removed that our study is the largest series lines 349 and 416 to 418.

- discussion: I would emphasize that your study confirms the daily clinical practice

We thank the reviewer for this comment. It is true that it should be noted that our results confirm daily clinical practice, namely that MMR/MSI status are concordant between primary tumor and metastases. This point has been added in the manuscript lines 356 to 358.

Reviewer 2 Report

This study is of value and seems technically well done, although I have remarks concerning quality and interpretation of MMR protein immunohistochemistry.

a) Figure 2, A2. You describe PMS2 positivity, however, the staining looks negative (picture bottom right)

b) Figure 2, B2. You describe MSH2 positive, however, the staining looks negative (picture lower left)

c) Figure 2, C2. You describe normal expression of MMR proteins. However, MLH1 (upper left) and PMS2 (bottom right) are negative. Therefore, this lung metastasis is MMR defect. If so, the lung metastasis is derived from the colon cancer from 2005 (it was MMR defect/MSI high). In my opinion the manuscript describes the results wrong and makes the wrong conclusion concerning this lung metastasis.

d) Figure 3, B2 down right. The PMS2 staining is of poor quality. Perhaps you can improve the figure by taking a photograph from an other area of the sample with a convincing PMS2 reactivity?

e) Materials and Methods. It would be an advantage to mention that the Pentaplex panel is from the company Promega (?)

Author Response

This study is of value and seems technically well done, although I have remarks concerning quality and interpretation of MMR protein immunohistochemistry.

 a) Figure 2, A2. You describe PMS2 positivity, however, the staining looks negative (picture bottom right)

Thank you for the careful advice. We have performed new immunohistochemistry in order to provide better images for all figures A2. We confirm that there is a positive but weak PMS2 expression in 400X magnification.

b) Figure 2, B2. You describe MSH2 positive, however, the staining looks negative (picture lower left)

In the same way we have provided better images for all figures B1 and B2 concerning the right colon cancer showing a MSH2 positive staining.

c) Figure 2, C2. You describe normal expression of MMR proteins. However, MLH1 (upper left) and PMS2 (bottom right) are negative. Therefore, this lung metastasis is MMR defect. If so, the lung metastasis is derived from the colon cancer from 2005 (it was MMR defect/MSI high). In my opinion the manuscript describes the results wrong and makes the wrong conclusion concerning this lung metastasis.

We have performed new immunohistochemistry and new pictures with a different device to highlight the expression of the 4 MMR proteins confirming the pMMR status of this lung metastasis.

d) Figure 3, B2 down right. The PMS2 staining is of poor quality. Perhaps you can improve the figure by taking a photograph from an other area of the sample with a convincing PMS2 reactivity?

We give a new picture for PMS2 staining with 300X magnification from another area in order to show a better PMS2 positivity.

e) Materials and Methods. It would be an advantage to mention that the Pentaplex panel is from the company Promega (?)

We have added that the Pentaplex panel is from the “Promega®” company lines 490.